# Three-dimensional Magnetic Induction Tomography: Improved Performance for the Center Regions inside a Low Conductive and Voluminous Body

**DOI:** 10.3390/s20051306

**Published:** 2020-02-28

**Authors:** Martin Klein, Daniel Erni, Dirk Rueter

**Affiliations:** 1Institute of Measurement Engineering and Sensor Technology, University of Applied Sciences Ruhr West, Mülheim an der Ruhr D-45407, Germany; martin.klein@hs-ruhrwest.de; 2General and Theoretical Electrical Engineering (ATE), Faculty of Engineering, University of Duisburg-Essen, and CENIDE–Center for Nanointegration Duisburg-Essen, Duisburg D-47048, Germany; daniel.erni@uni-due.de

**Keywords:** magnetic induction tomography (MIT), electromagnetic tomography, center sensitivity, 3D reconstruction, inverse problems

## Abstract

Magnetic induction tomography (MIT) is a contactless technique that is used to image the distribution of passive electromagnetic properties inside a voluminous body. However, the central area sensitivity (CAS) of this method is critically weak and blurred for a low conductive volume. This article analyzes this challenging issue, which inhibits even faint imaging of the central interior region of a body, and it suggests a remedy. The problem is expounded via two-dimensional (2D) and three-dimensional (3D) eddy current simulations with different transmitter geometries. On this basis, it is shown that a spatially undulating exciter coil can significantly improve the CAS by >20 dB. Consequently, the central region inside a low conductive voluminous object becomes clearly detectable above the noise floor, a fact which is also confirmed by practical measurements. The improved sensitivity map of the new arrangement is compared with maps of more typical circular MIT geometries. In conclusion, 3D MIT reconstructions are presented, and for the same incidence of noise, their performance is much better with the suggested improvement than that with a circular setup.

## 1. Introduction

In the field of biomedical engineering, magnetic induction tomography (MIT) attempts to map the conductivity of biological tissues, e.g., the human thorax, to evaluate deviations in lung density. In principle, MIT could be a quick, convenient, and harmless tomography method in comparison to magnetic resonance imaging (MRI) or computer tomography (CT). The blurred nature of the utilized induction fields and the ill-posed inverse problem are some of the significant challenges of using MIT [1]. However, even low resolution MIT could be useful for the localization of larger anomalies inside the human body to facilitate the earliest possible treatment. A quick, harmless and convenient whole-body tomography might be useful in both the medical and the security sectors. Unlike ultrasound, induction fields also permeate lungs, bones and gas-containing intestines; thus, all regions of the body are accessible. As an example of a security application, modern airport scanners for mass processing currently cannot detect non-metallic and dangerous or illegal materials inside the body. A specific issue of MIT is the poor central area sensitivity (CAS) of a low conducting voluminous body [2,3]; weak signal quantity and quality is obtained from the central region with respect to perturbations closer to the surfaces. From a practical standpoint, these signals are virtually useless even for the reconstruction of a faint image of the interior. Improved CAS promotes the reconstruction of the conductivity distribution throughout the volume, which is the topic of this work.

Typically, in the annular MIT setups that have been reported in previous studies—in which multiple transmitters and receiver coils are arranged around a low conductive saline cylinder [1,2,4,5,6,7,8,9]—the CAS becomes virtually zero [2,3] (Figure 1a,b). This problem is often bypassed by using a shallow conductive saline cylinder, where the inserted perturbation approaches the more sensitive top and/or bottom plane (quasi-2D [1,2,4,5], as seen in Figure 1d) or, alternatively, by only using the signals that originate from perturbations near the sensitive circumference [4,6,7,8,9] (Figure 1a–c). The signals are calculated using Equation (1), i.e., with the differential eddy currents in the conductive volume and the virtual vector potential of the receiver. The methods used to calculate the maps are described in Section 2.3 and Section 2.4. Only one excitation coil and two receiving coils at characteristic positions are shown here to demonstrate the general effects. 3D imaging requires many more coils around the conductive cylinder and at different heights [10]. However, a poor CAS remains a problem for all other pairs of transmitter and receiver loops.

More closely related to the MIT geometries discussed in this article, Igney et al. [11] described a planar array MIT with exciter and receiver coils in a gradiometric arrangement. Our recently reported MIT scanner [12] is still technically similar to the methods described therein, and an experiment (Figure 2) highlights the addressed problem with the central region of a body, qualitatively related to the weak CAS, shown in Figure 1. In this experiment, a test body travels linearly through an opposing and planar arrangement of a single exciter coil and a gradiometrically aligned receiver coil (parallel to the yz-plane, as seen in Figure 2a). The test body (Figure 2b) with dimensions related approximately to a human torso is a 33-litre saline bath (0.8 S/m) with an immersed 0.27-liter void (sphere with 0 S/m). The sphere is steady with respect to the travelling saline bath. Figure 2c presents the differential signals (the local sensitivity) obtained from the measurements with and without perturbation. The relative volume of the perturbation is 0.82% and higher than applied for the maps in Figure 1. For the general approach, the volume center (green lines), the face-centered position close to the lateral surface (red lines), and the face-centered positions toward the excitation (blue lines) and receiver (yellow lines) surface are the positions representing the void. The response from the center of the volume (green lines) is weaker than the other differential signals. The center signal is indistinguishable from the noise floor (black line). Thus, virtually no information is obtained from the center of the volume, in contrast to the other positions near the surfaces. It is important to note that the signal-to-noise ratio (SNR) for the measurement, shown in Figure 2c, already approaches a well-performing 60 dB, as further described in Section 2.1.

To date, none of the previously published MIT systems performed sufficiently for a 3D reconstruction of a low conducting and voluminous body, including the center regions. The following sections present the 2D and 3D eddy current simulations that were used to analyze the weak CAS in more detail and to derive the suggested improvement. The general strategy used in this work aims to show that a distinct suppression of disadvantageous eddy current fields can enhance the CAS. Such disadvantageous eddy current topologies occur from localized and discrete excitation loops that, typically, have been applied throughout the MIT literature. The proposed novelty is a more non-localized and quasi-infinite excitation field with a spatially sinusoidal modulation, i.e., the field of an undulating exciter coil.

## 2. Materials and Methods 

The magnetic induction principle is based on a primary magnetic excitation field that induces an electric field throughout a conductive test object where an eddy current field is established inside the conductive object. The eddy currents generate a weak secondary field, which is finally detected by the receiver coils. The primary flux in the receiving loop (Figure 2a) is almost completely suppressed by a gradiometric alignment; thus, the total signal (dotted black line in Figure 2c) exclusively represents the imprint from the secondary field of the eddy currents within the body. The phase shift is not considered; the signal amplitude of the secondary field can be measured directly for any phase angle, as recently demonstrated [12].

### 2.1. Characteristics of Limiting Noise

The quick measurements (10 s) are affected by random vibrations and displacements (mechanical noise), which add up to the differential signals, shown in Figure 2c. The overall system SNR for a single measurement approaches 60 dB: 1.3 V amplitude for the total signal (dotted black line) vs. 1.3 mV noise amplitude. Two subsequent measurements with identical conductivity distribution show a difference, due to the added noise energy, resulting in 1.8 mV (+3 dB, black line). Somewhat more noise occurs for the perturbation in the center (green lines), since a manipulated cuboid, not the identical cuboid, is repeatedly recorded. However, this is not representative of the noise of a single measurement. The 60 dB SNR for a biomedical approach is already well-performing compared to prior published MIT systems [8], and it is achieved with virtually perfect gradiometry and a relatively large and powerful excitation loop.

Noise that has no mechanical origin is not significant, for example, the electromagnetic interference (EMI) from other environmental sources or the Johnson-Nyquist noise in the receiving circuitry. The overall noise amplitude, with an origin other than a mechanical one, is <0.5 mV, and was determined by using empty measurements (i.e., signal fluctuations without mechanical scanning of a saline cuboid).

Mechanical noise originates from the scanner itself, and from the quasi-seismic vibrations of the building in which the laboratory is located. In a test, a living person causes even stronger “seismic” artifacts [3,12,13] than the mechanical noise. For a somewhat prolonged measurement (>1 s), better signals are not possible from living beings with even tiny mechanical activity, at the very least caused by an inevitable heartbeat [14]. Unlike EMI or Johnson noise, here the dominating dislocation noise is not simply an additive component. For example, a doubling of the excitation intensity, and to keep the signal within the linear range of the receiver, a halved amplification of the raw signal of the receiver, cannot improve the obtained level of dislocation noise. In contrast, the same procedure could decrease the electronic noise contribution in the final signals. Rather than being additive, the dislocation noise impinges in a relative or multiplicative manner. This is apparent for the green lines (center signal) and the black lines (noise signal) depicted in Figure 2c: The strongest noise amplitudes occur for those scan positions where the total signal of the cuboid (dotted black line) and its first derivative in the x-direction are strongest. The first derivative accounts for the signal fluctuations from dislocations or vibrations of the body in the x-direction. For dislocations in the z-direction, the total signal would fluctuate. Furthermore, a smaller cuboid or a cuboid with a reduced saline concentration would certainly result in a weaker total signal. However, the signal could be restored with a stronger excitation field; from a practical standpoint, it would result in the same noise level.

As the governing guideline toward a better CAS, the relative amplitude of the central differential signal must be increased with respect to the total signal of the cuboid. Therefore, the maps in Figure 1 are shown in relation to the total signal of the cuboid. An absolute increase in the total signal (e.g., with a stronger excitation field) would only proportionally increases the dominating dislocation noise. The quality of the differential signals would remain virtually unchanged. Moreover, as seen in Figure 2c, the full extent of the system´s measurement dynamic has already been exploited, and electronic noise is not the salient problem. Therefore, even receivers with a lower noise level, such as atomic magnetometers [15], would not improve measurements on living beings with natural movements.

### 2.2. Three Different Measurement Configurations

The detailed analysis and proposed improvement are presented via three different inductor geometries of the MIT scanner (Figure 3). In each of the three setups, the conductive body (Sheet or Volume) travels 256 cm in the x-direction through the gap between the exciter and the receiver. The receivers are gradiometrically aligned for the cases in Figure 3a,c. These cases, which are also presented experimentally, require a virtually perfect suppression of the primary signal in the receiver coils. Gradiometry is not required for the scenario in Figure 3b, since this setup is only theoretically examined.

As the received signal changes over the traveling coordinate x, a value is sampled every centimeter. Thus, a complete measurement signal is a curve consisting of 256 values, and the substantial imprints are obtained in the middle region (x = 53… 203), e.g., shown in Figure 2c.

In the first setup (Figure 3a), the exciter and receiver coils are circular, as typically found in the MIT literature. This setup was put into practice and the measurement results are shown in Figure 2c. In the second setup the exciter and receiver are vertically parallel wires (Figure 3b). An undulating exciter is used for the third scenario (Figure 3c), i.e., multiple equidistant wires with antiparallel current directions. As an additional feature, a butterfly coil (planar gradiometer [3]) is applied as a receiver. This receiver design better matches the secondary field of conductivity perturbations inside the volume, as further described below.

### 2.3. Mathematical Signal Calculation

The signal calculation (forward problem) for low conductive and extended bodies was analytically conducted in MATLAB (Version: R2019a), under the presupposition of weak-coupling [13], i.e., the induction fields at 1.5 MHz are barely altered by the low conductivities within the volume [16].

The primary field induces an electric field $E_{E}$ throughout the body, which is proportional to the vector potential $A_{E}$ ($E_{E}=-j\omega A_{E}$) of the exciter coil. Analytical expressions for circular currents [17,18] or segments of straight wires are used for the vector potential of the exciter and the receiver, which can be arbitrarily superimposed to, e.g., rectangular inductors. The formulations are presented in Appendix B. With the calculated vector potential $A_{E}$ and an assumed conductivity distribution σ within the volume V, the 3D eddy current density J inside this volume can be numerically computed, as described in [14,19]. The reciprocity theorem [17,20] states that the inner product between the vector potential $A_{R}$ of a receiving inductor and the eddy current density J in the volume V represents the received signal S:(1)$$S\sim \overset{\;}{\underset{V}{\int }}J\cdot A_{R}\;dv$$

For the planar scanners (Figure 3), a different J occurs for each x-position of the body inside the exciter vector potential $A_{E}$, and must be individually computed (J as a function of midpoint x-position $x_{m}$ of the body =$\;J_{x_{m}}$). This particular calculation creates a major contribution towards the overall computational effort. As a fortunate coincidence, an undulating exciter offers an efficient shortcut (Section 2.3.1) for the frequently required calculation of $J_{x_{m}}$. The proportionality in Equation (1) is sufficient, since the relative amplitudes of the signals and differential signals are of particular interest in this article, not the absolute amplitudes.

#### 2.3.1. Special Case-Undulating Exciter

On the assumption that there is a quasi-infinite undulating exciter, the laborious calculation of $J_{x_{m}}$ can be accelerated. Close to the exciter wires and along the x-direction, an almost rectangular vector potential is obtained, with the basic spatial frequency of the distance (D in Figure 4) between two wires with the same current direction. The higher spatial frequency components decrease more strongly in the z-direction than the basic frequency; thus, only a sinusoidal vector potential exists in the measuring range. Due to the linearity of the eddy current solution in a sinusoidal vector potential, it is sufficient to calculate the currents for two positions instead of calculating 256 positions. Let the eddy current solution $J_{x_{m}}$ in front of a wire (position $x_{0}$, Figure 4) be $J_{\Phi }$. When shifting the body from $x_{0}$ to $x_{0}+\frac{D}{4}$, the position between two wires, the obtained eddy current solution will become $J_{\Psi }$. For any arbitrary position $x_{m}$ (c.f. Figure A2, Figure A3 and Figure A4 in the appendix) of the body, the general solution is simplified as follows:
(2)$$J_{x_{m}}=J_{\Phi }\cos\left(\frac{2\pi \Delta x}{D}\right)+J_{\Psi }\sin\left(\frac{2\pi \Delta x}{D}\right)$$
with $\Delta x=x_{m}-x_{0}$. Only two eddy current solutions ($J_{\Phi }$ and $J_{\Psi }$) have to be calculated, and their weighted superposition reveals $J_{x_{m}}$. More detailed insights regarding Equation (2) are presented in Appendix D.

### 2.4. Differential Signal, Sensitivity Matrix, and Sensitivity Map

In measurements within a conductive background (e.g., Figure 2b), a differential signal ΔS is obtained by subtracting a measurement signal with perturbation from one without perturbation. Therefore, this ΔS indicates the sensitivity to a local conductivity change in a generally conducting background.

The sensitivity matrix contains the differential signals for each voxel inside the volume [21]. For example, if only applying one receiver, 256 measurement steps and 100 voxels inside the object, a matrix with 256 rows and 100 columns is obtained.

The sensitivity maps (e.g., in Figure 1) show the relation between the differential signal and the absolute signal in percentages. This ratio weighs the robustness of the sensitivity against the dislocation noise, as described in the following subsection.

### 2.5. Iterative Image Reconstruction

An iterative algorithm is used to reconstruct the 3D conductivity distribution inside a voluminous body using the following procedure:An “unknown” body with an inhomogeneous conductivity distribution is measured (here, calculated via the forward problem) and results in the “real” signal $S_{R}$. Noise is added to the computed signal with the SNR being set to 50 dB; thus, it is 10 dB lower than the practical SNR, shown in Figure 2.An estimated body with the same outer contour, but with a homogenous conductivity distribution is calculated, and results in $S_{E}$.For subsequent and iteratively corrected body estimations, the total signal difference ($\Delta S_{T}=S_{R}-S_{E}$) should be minimized (i.e., minimizing the root mean square [RMS] of $\Delta S_{T}$), ideally down to the noise level. Therefore, the homogenous conductivity distribution $\sigma_{E}\left(x,y,z\right)$ inside the estimated body is locally increased or decreased by small amounts to maintain the stability of the iterative procedure within certain limits (e.g., the conductivity cannot become negative, and it also cannot exceed specific values for biological tissue), based on the inverted and regularized sensitivity matrix (Jacobian K) and the differential signal $\Delta S_{T}$ [21].The corrected $\sigma_{E}\left(x,y,z\right)$ leads to a new $S_{E}$, which in turn leads to a generally decreased RMS of the new differential signal $\Delta S_{T}$. A new K must be calculated for the modified conductivity distribution, and further correction of the estimated body occurs with the current $\Delta S_{T}$.The overall procedure is repeated until no further decrease of $\Delta S_{T}$ can be obtained. Then, the last modified conductivity distribution should approach the unknown conductivity distribution of the real body, within the general limits of the ill-posed MIT principle.

An iterative procedure is required, since, due to the spatially-extended eddy current fields, the signal imprint from a local perturbation also depends on the global conductivity distribution, which is also unknown. A direct solution in a single step is hampered by the great complexity of the generally non-linear MIT problem. The inverted sensitivity matrix $K^{-1}$ is regularized by the iterative Landweber method [22,23,24], and used for linear and stepwise approximations towards the reconstruction of the unknown body.

## 3. Analysis of Three Different Exciter Setups and Their Effect on the Eddy Current Distribution and the Obtained Sensitivity

### 3.1. Eddy Current Distribution in a Travelling Conductive Sheet (2D Object)

It is instructive to analyze the case of a travelling, vertically-oriented, conductive sheet in the middle plane between the exciter and receiver coils (see the sheet in Figure 3). The intentionally chosen middle plane (axial distance to the coils = 20 cm) is the most disadvantageous placement, since both fields, the exciting field $A_{E}$ and the receiving field $A_{R}$, are widened and diffused; no sharp response for the features can be expected here, in contrast to the positions either near the exciter coil or the receiver coil. The 2D eddy currents can be clearly visualized to clarify the underlying problems. The seemingly trivial 2D calculations, with only two representative positions for a local inhomogeneity, provide a clear demonstration of the improvements.

The left side of Figure 5 illustrates the eddy current density J (blue arrows) for a conductive sheet with height 40 cm and width 40 cm seen from the receiver direction. The figures show the sheet in different x-positions (A, B, and C) in front of the exciter. The regions with no eddy current are labelled with Z (Zero). Local conductivity voids (perturbations) are introduced in the sheet (middle height y = 20 cm), at the lateral edge (red quadrant) and in the center (green quadrant), which results in differential signal amplitudes (red and green lines on the right side) with respect to a complete sheet without voids.

#### 3.1.1. Conductive Sheet in a Circular Coil Setup

This experiment begins with a circular setup (Figure 5). The setup consists of an exciter with a large excitation loop (diameter 40 cm) with high range and a smaller receiving loop (diameter 10 cm) in gradiometric alignment (Figure 3a). The axial distance (z-distance) between the two coils is 40 cm. A parallel aligned receiver coil is not considered here, as this leads to an even lower CAS (≈1/750, see Appendix F) and the primary flux through the receiver loop would not vanish (no gradiometry) in practical measurements. Smaller loops, as typically applied to circular MIT arrangements, would barely change the outcome, while the fields from these loops widen over a distance of 20 cm in the z-direction, ultimately leading to very similar results.

For positions A, B, and C, the current densities in the center region are weak, or even zero; the strongest currents tend to flow in a circumferential direction along the edges of the sheet. The final resulting differential signals are much stronger for a void at the edge (red line) than for a void in the center (green line). Thus, the sensitivity at the edge position is much higher than the CAS (Table 1).

The circumferential eddy current density, shown in Figure 5, position C has an O-shaped eddy current distribution and is denoted as the O-mode. With maximum current densities along the edges, it is the dominant solution (relatively strong), and it has only one zero area (Z) at or near the center of the sheet; this zero area never occurs at the edges. This behavior regularly results in a poor CAS.

Figure 6 illustrates the shape of a differential eddy current density, i.e., the subtracted current field densities from a sheet with and without perturbation. Due to inhomogeneity in a homogeneous environment, the shape of the differential eddy current density approaches a dipole-like field (Figure 6a). However, the edge of the sheet distorts this field (Figure 6b). The two eddy current perturbations in the sheet generally appear in this shape, and only the intensity and/or the sign changes for the momentarily applied symmetry in the vertical direction.

#### 3.1.2. Conductive Sheet in a Vertical Wire Setup

As seen in Figure 7, the exciter and opposing receiver are vertically arranged wires (Figure 3b), again at a z-distance of 40 cm. The length in the y-direction is 50 cm. The differential signal of a perturbation in the middle (green line) becomes relatively stronger with respect to that of the edge (red line). The sensitivity in the middle region (Table 2) is higher with respect to the information presented in Figure 5 or Table 1; it is particularly enhanced, due to the effects occurring in position C.

Looking at this symmetrical position in front of the wire (position C), a Φ-shaped eddy current distribution (Φ-mode) is enforced by symmetry. Now having two zeroes and all the excited current passing through the center region, the Φ-mode enhances the CAS.

When moving the sheet out of symmetrical position C (Figure 7, position A and position B), a circumferential eddy current shape (O-mode similar to the one presented in Figure 5, position C) is re-established. Note that the total signal (dotted black line in Figure 7) is weaker for position C than for position B, because the overall coupling of the sheet via O-mode is more effective than via Φ-mode.

The evolution of the suggested improvements, which will finally also apply to 3D bodies, is described as follows:When tracing the current values of the O-mode (Figure 5, position C) along the midline (x-direction) at y= 20 cm, the result would be similar to half a period of a sinusoidal signal. The O-mode corresponds to a spatial frequency in the x-direction, and the lateral size of the sheet equals half a period of this spatial frequency.When tracing the current values of the Φ-mode (Figure 7, position C) along the x-direction, the result would be similar to a full period of a sinusoidal signal, i.e., a doubled spatial frequency in the x-direction with respect to the O-mode.

Due to the linearity of the excited eddy current densities, it can be stated, in essence, that a distinct suppression of specific low spatial frequencies in the excitation field and in the x-direction can inhibit the disadvantageous and stronger O-mode for every scan position x. In other words, the O-mode can be eliminated via destructive interference, and the primary field exclusively excites the eddy current fields with more than one zero. Thereby, on average, there is a relative increase in the number of eddies for the center region.

#### 3.1.3. Conductive Sheet in a Undulator Exciter Setup

To achieve such a distinct suppression of the O-mode, more vertical wires with alternating current directions are applied as the exciter (Figure 3c), i.e., an undulator. The y-dimension of the wires is 50 cm, and 11 parallel vertical wires in a row are arranged in the xy-plane. The periodicity *D* of the undulator is chosen to match the Φ-mode in the sheet (Figure 7, position C). As an additional feature in Figure 8, a butterfly coil (x-length = 20 cm and y-height = 10 cm) is applied as a receiver; while its virtual vector potential $A_{R}$ better adapts to the dipole-like current field at the center (Figure 6a), it less effectively matches the edge distorted dipole (Figure 6b). The undulator exciter also promotes the CAS with a circular receiver coil (Appendix G), but a butterfly receiver fits better to the secondary field of a central perturbation.

The simulated outcome of the information, shown in Figure 8 and Table 3, is that the center signal (green line) better approaches the amplitude of the outermost lateral void (red line) than the other setups. Moreover, the CAS is increased by more than factor 6 of magnitude compared to Table 1. Note, the frequencies of the signal in the x-direction are apparently higher in Figure 8 than in Figure 5 and Figure 7, which indicates better localization capabilities [5] for the MIT reconstruction.

In addition to the Φ-mode (Figure 8, position C), the previous O-mode is replaced by an eddy current with a thinner O-shape (Ψ-mode, position A) with three principal zeroes, one in the middle and two at the lateral edge positions. The Φ-mode and Ψ-mode correspond to the spatial periodicity of the undulator. In contrast to the absent O-mode, the Ψ-mode is not sensitive to a deviation at the lateral edges. For the Ψ-mode, the trace of the current amplitudes at the midline (y= 20 cm) would be similar to one period of a sine function, and the Φ-mode would be similar to one period of a cosine function. Moreover, enforced only by symmetry, the Φ-mode and Ψ-mode are orthogonal. These two current fields already provide a complete basis for all other J(x), for example, the J in the intermediate position B could be exactly matched with an appropriate superposition of the Φ-mode ($J_{\Phi }$) and Ψ-mode ($J_{\Psi }$). Accordingly, when adding the current density values of the two basic fields ($J=\sqrt{J_{\Phi }{}^{2}+J_{\Psi }{}^{2}\;}$), the resulting J would be almost homogeneous at the mid-height of the sheet (y = 10…30 cm). On average, the eddy currents for the center and at the lateral edges are nearly the same. In contrast, the average currents, shown in Figure 5 and Figure 7, are distinctly weaker for the center than for the lateral edges.

Before proceeding to a detailed discussion of the 3D setup, a brief overview of the development of the 2D scenario is provided below.

#### 3.1.4. Intermediate Discussion (2D)

Three different coil geometries have been discussed for a planar MIT scanner. Among these three setups (Figure 5, Figure 7 and Figure 8), the CAS, an important measure in this research, has been improved by more than 15 dB, from about 1/305 (Table 1) to over 1/126 (Table 2) and 1/50 (Table 3). Admittedly, the total signal strength (dotted black line) has significantly decreased from 0.435 (Table 1) to $6\times 10^{-3}$ (Table 3). This major decrease can actually be compensated for with more current in the exciter and increased gain in the receivers, as shown below in practical experiment. Note that the relative decrease in the differential signal of a perturbation in the center is less than the decrease in the total signal. Finally, the differential signal of the central area will be lifted significantly above the noise floor. Furthermore, the Φ-mode and Ψ-mode, shown in Figure 8, carry less current density than the O-mode, shown in Figure 5. Thus, for the same power irradiation, currents that are several times stronger can be applied in the undulator, partly compensating for the loss. For these reasons, it is now anticipated that, in the following 3D presentations, the total signals approach the same amplitude for all of the different MIT setups.

Everything can be called into question by the argument that the exposed effects could also be obtained by an afterwards and numerical allocation from many and individually controllable coils, as are usually present in current MIT systems. Thus, everything is already available within the prior art. An extended and dense array of circular coils could indeed reproduce the undulator´s field. An exact matching (in amplitude and phase) of the individual currents for the multiple coils is however technically demanding. Moreover, another current pattern than that for the spatially prefiltered undulator field (where the low spatial frequencies in x-direction are absent) is not helpful for the CAS. The direct realization of a large, hardwired, powerful and stable (low noise and drift) undulator is technically better feasible than an extended coil array with individual controls.

### 3.2. Eddy Current Distribution in a Travelling Conductive Volume (3D Object)

For the 3D calculations, a homogeneously conducting cuboid was used with only one relatively small inhomogeneity (a non-conducting void) at certain characteristic positions: The volume center position, the face-centered position at the lateral surface, the face-centered position towards the excitation surface and the face-centered position towards the receiving surface. The dimensions of the computed cuboid and the void are similar to the dimensions of the practical saline body, shown in Figure 2b, i.e., the dimension of the calculated body is representative for a biomedical application (e.g., a human torso).

Detailed images from the 3D eddy current fields are not very descriptive. The cross-sectional view of the horizontal middle plane of the cuboid is more instructive (left sides of Figure 9, Figure 10 and Figure 11). The volume center and the face-centered positions are visible, and all the x- and z-components of J vanish, due to vertical symmetry (as seen at the mid-height [y=20 cm] in Figure 5, Figure 7 and Figure 8). The remaining $J_{y}$ is displayed as a plot of color-coded intensity with a sign shown on the cross-section.

#### 3.2.1. 3D Object in a Circular Coil Setup

The applied geometry, shown in Figure 9, is closely related to the practical measurement depicted in Figure 2. It shows the travelling cuboid between the large excitation loop with high range and a smaller receiving loop (diameter 10 cm) in gradiometric alignment (see also Figure 3a). The calculated signals (Figure 9, right side) resemble the practical measurements, shown in Figure 2c. The eddy current distribution is disadvantageous, because $J_{y}$ obtains only one zero area (marked with “Z”), which tends to cut the cuboid´s middle region for all scan positions. The relative amplitude of a central void accounts for less than 0.1% of the total signal (Table 4) and is readily submerged under even moderate noise (e.g., the 60 dB SNR apparent in Figure 2c).

The scenario is inherently related to the typical MIT arrangements with circular coils around a measurement volume that have been frequently reported [1,2,4,5,6,7,8,9], which either only includes quasi-2D results or are restricted to perturbations near the surface of the volume (Chapter 1, Paragraph 3). Scharfetter et al. [23] have mentioned that a symmetrical arrangement with circular coils strongly discriminates against the center position.

#### 3.2.2. 3D Object in a Vertical Wire Setup

The setup, shown in Figure 10, corresponds to the setup, shown in Figure 7, with two vertical wires as the exciter and receiver, respectively (see also Figure 3b). As a disadvantageous setback with respect to the 2D sheet, even for the symmetrical x-position C, no 3D version of the Φ-mode appears at depth within the body. Instead, one single zero area predominates in the middle regions, and the relative signal amplitude from the center is only slightly improved (it is still only about 0.3%, Table 5). This is also the case for the other scan positions.

#### 3.2.3. 3D Object in an Undulator Setup

More significant improvement (Figure 11 and Table 6) was seen in the setup with an undulating exciter and a gradiometrically-aligned butterfly coil used to receive signals (as seen in Figure 8). Principally, due to the absence of low spatial frequencies in the excitation field, more than one zero is enforced in the cuboid, and the 3D versions of the Φ-mode and Ψ-mode are apparent (position C and position A, respectively). The eddy currents. shown in Figure 11, could actually be obtained via appropriate superposition of the various eddy currents in the arrangement presented in Figure 10. Now, however, constructive interference is obtained for the center currents. In particular, no vanishing eddy current density is present for the central area in Figure 11, position C, in contrast to Figure 10, position C. Practically relevant is that the helpful effects are not just restricted to cuboids with a homogeneous background conductivity, but still effective for more arbitrarily shaped bodies with heterogeneous conductivity (c.f. Appendix E).

To better compare the undulator setup and the frequently published circular MIT arrangements (Figure 1), the undulator sensitivity map for the horizontal middle plane of a conductive volume is presented in Figure 12a. For faster calculations (Section 2.3.1), a quasi-infinite undulator was used. Instead of only calculating a few characteristic positions within the plane (Figure 11), all the positions are calculated. Since every point with a local deviation Δσ results in a differential signal $\Delta S\left(x_{m}\right)$, and not just a single measurement value, the signals’ RMS values are shown as a representation for each point (Figure 12a). The relative volume of the perturbation is smaller than in Figure 7, Figure 9, and Figure 10; it matches the relative volume for the maps in Figure 1.

The same arrangements are shown after high-pass filtering (spatial differentiation) of the related $\Delta S\left(x_{m}\right)$ (Figure 12b). This high-pass filtering weights the signal value for localization; weak localization capability (i.e., blurred response) is provided from a low frequency or even a direct current (DC) signal. At the other end of the scale, a Dirac-like pulse with the same RMS at a certain position $x_{m}$, with the highest inherent frequency components, would provide the most valuable information regarding the position of the perturbation. The values, shown in Figure 12, are normalized to the total RMS signal of the body (in Figure 12b also high-pass filtered). Therefore, they are directly comparable to the maps, shown in Figure 1. The local weaknesses at 10 and 40 cm (Figure 12) result from the generally weak eddy currents in these regions (caused by the lateral boundaries), because both of the eddy current distributions – the Φ-mode and Ψ-mode in Figure 11 — are not strong in these particular locations. However, the local sensitivity decreases are much less detrimental for the 3D reconstructions than a more general and weaker CAS. In direct comparison with the maps of the elongated cylinder, shown in Figure 1, the information presented in Figure 12 suggests clear superiority of the undulator enhanced MIT scanner.

By comparison, Figure 1 illustrates some well-known sensitivity maps [2,3], depicting a more typical, circular MIT setup. As seen, very satisfactory sensitivity was obtained for the upper (or lower) level of the shallow bath (Figure 1d). However, this quasi-2D scenario only works for perturbations that contact the upper or lower level of the shallow bath – the eddy current densities concentrate near those surfaces. For much more elongated cylinders (Figure 1a,b), approximately related to the volume used in the present study (Figure 12), the situation becomes even more disadvantageous. The CAS (Figure 1a) is virtually zero, and it remains very poor for any of the horizontal planes above or below the center plane (Figure 1b). However, the strongest responses occur in the cylinder´s surface regions. Small dislocations would result in relatively large artifacts. Thus, this classical MIT arrangement is not very suitable for voluminous objects.

#### 3.2.4. Intermediate Discussion (3D)

For the signals received in the undulator setup (Table 6), the relative amplitude of the signal from the central region is enhanced with respect to the more superficial positions near the surface, and particularly with respect to the total signal of the cuboid. Here, a relative amplitude of about 2.2% (1/45) was obtained, i.e., > 27 dB with respect to the circular setup (Figure 9) and > 18 dB with respect to the vertical wire setup (Figure 10). The essential improvement of the CAS was achieved by the undulator exciter. Even an undulator exciter with a circular receiver (Appendix H, Table A3) improves the CAS by about a factor of ten (≈20 dB) compared to a circular coil setup (Table 4). The main reason for this is the non-vanishing eddy current in the center region (Figure 11, position C), only the undulating excitation field could achieve this in the 3D body. The butterfly receiver further enhances (ca. 6 dB) the response for a centered perturbation (dipole shaped field), but cannot compensate for weak, or even vanishing eddy currents in the center of the volume, as they occur with other excitation fields (Figure 9 and Figure 10).

Further fine-tuning would be possible using a slightly-adapted periodicity of the undulator and modified geometry of the butterfly receiver. A smaller spatial period *D* of the undulator (Figure 4) further increases the CAS. Conversely, a smaller period directly reduces the extent of the field in the z-direction and strongly reduces all the signal amplitudes, i.e., ultimately, electronic noise or EMI will become a limiting issue. Here, the calculated undulator is intentionally finite with just five wires being used to better match the undulator implemented in practice in the subsequent experiment (Section 3.3). Such a finite undulator cannot ease the forward problem, so Equation (2) does not apply. Nevertheless, the intended topology of the projected field is already emphasized to prefer excitation of the Φ-mode and Ψ-mode. The field from this non-infinite undulator is presented in Appendix C.

As shown by comparing the undulator setup (Figure 12) with the classical MIT setup (Figure 1), the undulator is much better adapted for the detection of central perturbations in a volume.

### 3.3. Experimental Verification

Figure 13 shows the undulator, a gradiometric butterfly receiver, and the measurements from the same saline cuboid with a sphere (cf. Figure 2b) at the four characteristic positions. For technical reasons, the undulator is not infinitely wide. However, as also applied in the computational results presented in Figure 11, it preliminarily consists of five copper strips with antiparallel currents. The adjustment of a balanced current distribution (in amplitude and phase) is not trivial [12]. The first and the fifth wire are considerably thinner, as they should carry only half of the current. As sketched in Figure 13a, the wires of the undulator are driven with only one voltage source (10 V amplitude at 1.5 MHz), which directly translates into an equal electric field over the lengths of the strips. In the area near the vertical strip, the initially given and fixed electric field E is directly proportional to the projected vector potential (E=−jωA). This method appears to be much more convenient and stable than a method that drives the strips with individual voltage or current sources, which then must be carefully adjusted in amplitude and phase.

The measured signals, shown in Figure 13c, are closely related to the computational results presented in Figure 11, although not exactly the same geometries apply for all the components. For the volume center, the SNR, with respect to Figure 2c, is significantly increased by at least 20 dB, for the same incidence of vibrations, dislocations, or any other noise or drift, and without averaging or filtering the quick measurements (10 s). The center signal amplitude (green lines) is about one order of magnitude stronger than the noise (black line, still accounting for −60 dB with respect to the total signal). In accordance with Figure 11, the central relative amplitude approaches 2%, and thus, it is even higher than the relative volume of the non-conducting sphere (0.82%). Furthermore, the spatial response in the horizontal middle plane becomes more uniform; the various differential signal amplitudes are similar. The signals clearly deviate in shape, thereby distinguishing the z- and x-positions of the sphere. It can be reasonably concluded that the theoretical considerations from the previous section are essentially confirmed.

Vertical information (y-direction) from the saline cuboid is not addressed in this experiment, but it would be accessible via several more receivers at different heights, as applied to the reconstructions discussed in the next section. Vertically deviating features would approach the upper and lower surfaces of the cuboid, where, in general, the current density J is not critically small (Figure 1d).

## 4. Three-dimensional Reconstruction of Low Conducting and Voluminous Bodies

This section demonstrates the advantage of the improved CAS for a 3D reconstruction of voluminous bodies with an inhomogeneous interior, similar to the saline cuboid, shown in Figure 2. To utilize the shortcut from the forward problem of a quasi-infinite undulator (Section 2.3.1), an undulator with 11 current strips was applied. Figure 14a shows the total signals of the cuboids when travelling along an ideal and infinite undulator and along a more realistic and finite undulator with 11 wires (field shown in Appendix C). The total signals (Figure 14a, top) are very similar, however, they are not identical, i.e., the value of the deviation (max. 0.045) approaches the imprint of a local conductivity deviation, so it is already misleading. Nevertheless, the differential signals (Figure 14a, bottom), due to the local conductivity perturbation, as required for the much more computer-intensive Jacobian K, are sufficiently similar (max. deviation 0.003) for the two cases. Thus, the shortcut from the forward problem via the idealized Φ-mode and Ψ-mode, and Equation 2.2 can be exploited for the “real” undulator with 11 wires.

Figure 14b shows two setups for the 3D reconstruction (as described in Section 2.5), one with a circular setup and one with an undulator setup (11 current strips). Six butterfly receivers for y-discrimination were applied for both scenarios: Three smaller butterflies in a vertical column and — for the undulator at the next gradiometric position — three wider butterflies in a vertical column. The sensitivity of the smaller butterflies decreases faster over the z-distance than the sensitivity of the wider butterflies, thereby finally improving the depth resolution in the z-direction. To create a realistic state, random noise is added to the computed “real” signals ($S_{R}$), and a decreased SNR of 50 dB instead of the practically obtained 60 dB (Figure 2c and Figure 13c) is applied. The 50 dB SNR better approaches the characteristic dislocation noise from living beings [12]. The six independent receiver signals are combined into relatively long signal vectors with six distinct signatures. The initial $\Delta S_{T}$ with the first body estimation (a homogeneously conducting cuboid) and the final $\Delta S_{T}$ after reconstruction are shown. Note that, relative to the signal, more noise is apparent for the circular exciter, although the same SNR was adjusted for the “real” signal $S_{R}$. The applied algorithm is not capable of further decreasing the $\Delta S_{T}$ of the circular exciter, whereas $\Delta S_{T}$ can be better minimized with the undulator, resulting in better convergence. The left side of Figure 14b shows the “real” conductivity setup. The coloration mirrors the conductivity: Transparent = 0.5 S/m (the conducting background), blue and no transparency = 0 S/m, orange and no transparency = 1 S/m. With the same algorithm, the 3D reconstruction obviously performs much better for the undulator than for the circular exciter. For the other conductivity distributions inside the volume and their 3D reconstructions, the undulator remains clearly superior (Figure 14c,d).

The 3D body is discretized into 455 voxels. Thereby, each voxel accounts for 0.22% of the total volume of the cuboid. Particularly for the interior of the body, it is not practical to expect that features that are considerably smaller than 1% of the volume will be sufficiently detected (Figure 13), i.e., a much larger number of voxels would not help. Moreover, to cope with given computational resources a rather limited number of voxels is used, since the number of eddy current calculations frequently required for the iteratively changing Jacobian K and the number of inversions disproportionally increase with the number of voxels.

Note that here, with the number of voxels (455) and the length of the signal vector (6×200=1200), the applicable Jacobian already amounts to 546,000 elements. In terms of practical relevance, the undulator-enhanced reconstructions in Figure 14 are computed in 30 s with a standard office computer without accelerating hardware, such as a graphic processing unit (GPU) [25], and thus, the processing time already approaches the practical measurement time of the scanner (10 s). For the simple loop, due to the much more demanding forward problem without the accelerated undulator calculation (Section 2.3), the overall computation takes 5 min. Thus, the overall computational effort is substantially reduced to 10% with the undulator-based shortcut in the forward problem (Section 2.3.1).

## 5. Conclusions

Circular exciter coils, which are typically used for MIT arrangements, induce circular eddy current distributions with the highest current densities near the surfaces of a conductive object. Thus, a central region with vanishing currents is established so that the CAS becomes critically weak. As an improvement, the undulating excitation setup presented in this article does not result in vanishing eddy current densities in the central region of a volume, and a gain of more than 20 dB is achieved for the CAS. Importantly, for the center of the volume, the ratio between the differential signal and the total signal is experimentally shown to be higher than the ratio between the volume of the perturbation and that of the whole body. The shapes of the signals clearly deviate (i.e., the positions are distinguishable) and even bear higher frequencies (i.e., they provide more localization capability) than from circular and simple exciter coils.

Furthermore, an infinite undulator with only one distinct spatial frequency in the x-direction can significantly ease the forward problem; only two principal eddy currents must be calculated. For the computer-intensive Jacobian, a practically “infinite” undulator, which ends at greater lateral distances (x-direction) to the receivers, is already sufficient as there is almost no signal contribution from those distances, e.g., using 11 wires, as calculated for the 3D reconstructions, and shown in Appendix C.

Due to these improvements, reconstructions with the undulator setup are much more accurate than reconstructions with a round transmitter setup, especially in the central area of an object (c.f. Figure 14).

The following conditions should be considered for the dimensions of the scanner setup and for the object being scanned (c.f. Figure 3c). If the scanning path is long enough in the x-direction, the body can be arbitrarily wide in that direction. In the y-direction, the undulator should be twice the height of the object. The ideal test object is thin in the z-direction, and accordingly, it allows for a small gap between the exciter and receiver, thereby enabling an undulator preferably with low periodicity D (c.f. Chapter 3.2.4, Paragraph 2). A smaller periodicity D of the undulation further increases the resolution. Conversely, smaller periodicities directly reduce the reach of the field in the z-direction, and they strongly reduce all the signal amplitudes such that the electronic noise or EMI will ultimately become a limiting issue. Presumably, a longitudinal scan of a person lying down (Figure 15, flattened by gravity and thereby lowering the z-extension) is more suitable than a scan of a person in an upright standing position [12]. In addition, the torso of a person lying down has similar dimensions as the previously simulated and measured saline bodies. Moreover, lying down could further reduce motion artifacts from a person, and some receivers could be placed in lateral body positions to better approach the volume. Ultimately, the motion artifacts of a person lying down and with lateral stabilization will be smaller than the artifacts of the person in an upright standing position.

The undulator enhanced reconstructions promise more success for the interiors of living beings with a decreased SNR, i.e., 50 dB [12] instead of the practically apparent SNR of 60 dB (Figure 2 and Figure 13) from the saline cuboid. Although many demanding tasks remain to be solved, the development of a quick and convenient 3D MIT is getting closer to being realized.

## Figures and Tables

**Figure 1 sensors-20-01306-f001:**
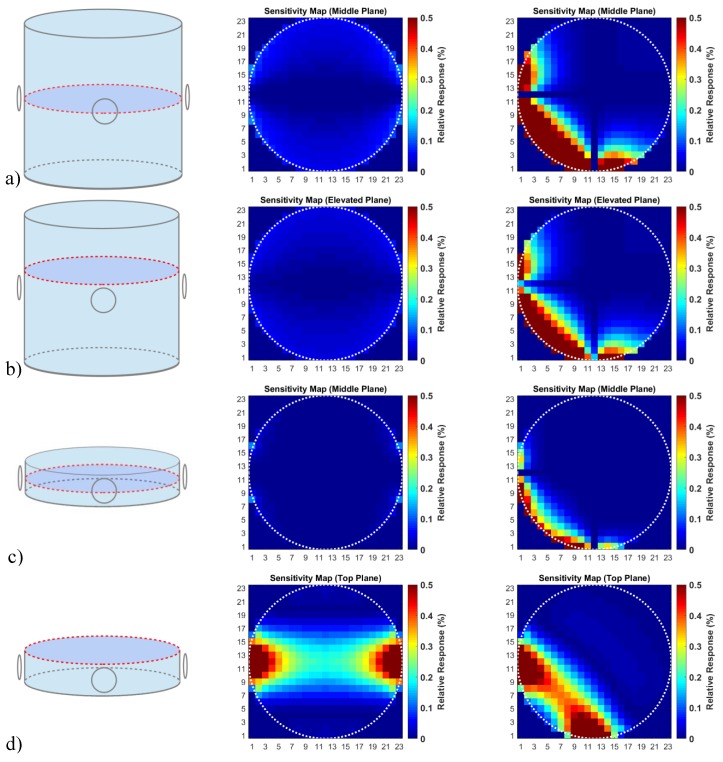
Circular MIT system with exciter and receiver coils in two typical alignments, i.e., in opposing and rectangular placements (left column). Middle and right column: Calculated sensitivity maps, the signal deviations (sensitivity), due to a local conductivity change are shown in relation to the total signal from the conductive volume. Middle column: Opposing coil placements. Right column: Rectangular coil placements. (**a**) Horizontal middle plane of an elongated saline cylinder; (**b**) vertically elevated plane; (**c**) the horizontal middle plane of a shallow bath is virtually undetectable; (**d**) the top plane of a shallow saline cylinder demonstrates very satisfactory detectivity throughout the plane.

**Figure 2 sensors-20-01306-f002:**
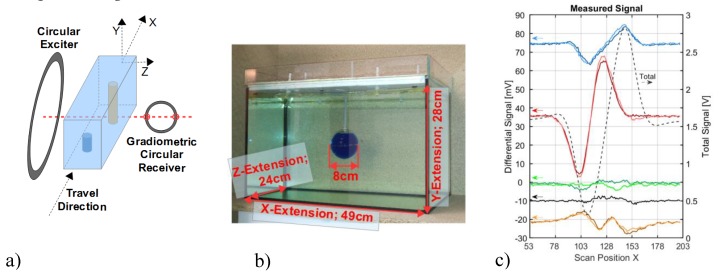
Experiment with a planar MIT setup. (**a**) Schematic illustration; the whole saline body travels linearly in the x-direction through a gradiometric arrangement of excitation and receiving loop. (**b**) The 33-liter saline body with a non-conducting 0.27-liter sphere in the horizontal middle plane. The sphere is steady with respect to the travelling saline body. (**c**) Measurements: The dotted black line represents the total signal of the cuboid; the red lines are the differential signals of the sphere when face-centered at the side wall; the blue and yellow lines represent the signature of the sphere towards the exciter and receiver coils (face-centered). The green lines from the center of the volume are weak, and they approach the noise floor (black line).

**Figure 3 sensors-20-01306-f003:**
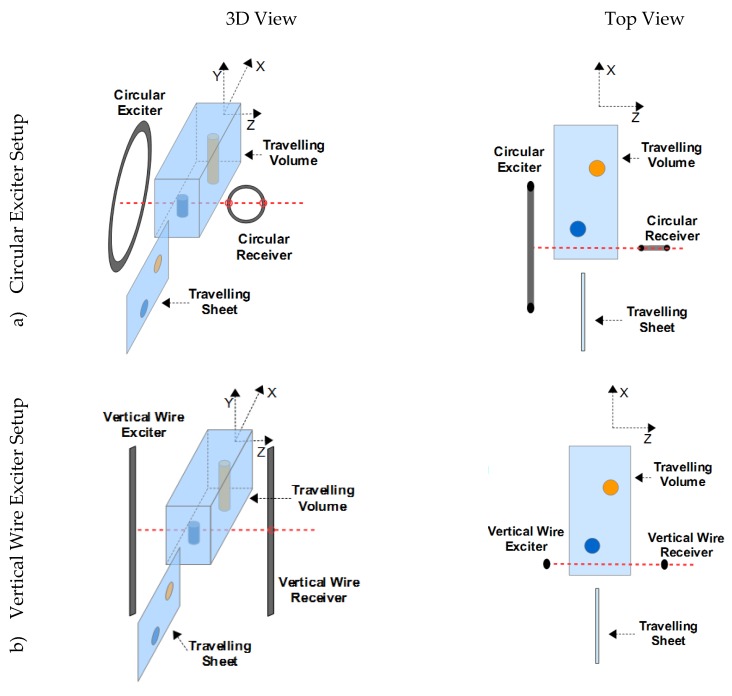
The 3D and the top view of three different kinds of planar MIT setups are shown. A sheet-like or a volumetric object travels in the x-direction through the arrangement. (**a**) Circular exciter setup; (**b**) vertical wire exciter setup; (**c**) undulating exciter (undulator) setup.

**Figure 4 sensors-20-01306-f004:**
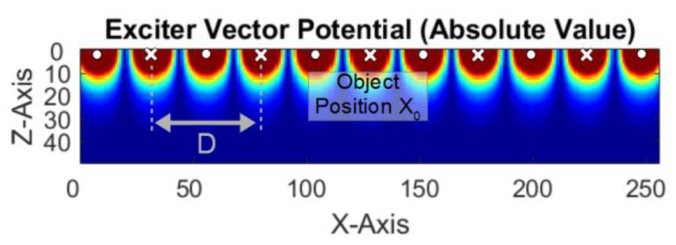
Top view of the vector potential ($A_{E}$) from a quasi-infinite undulating exciter; the white dots and crosses symbolize the alternating current directions.

**Figure 5 sensors-20-01306-f005:**
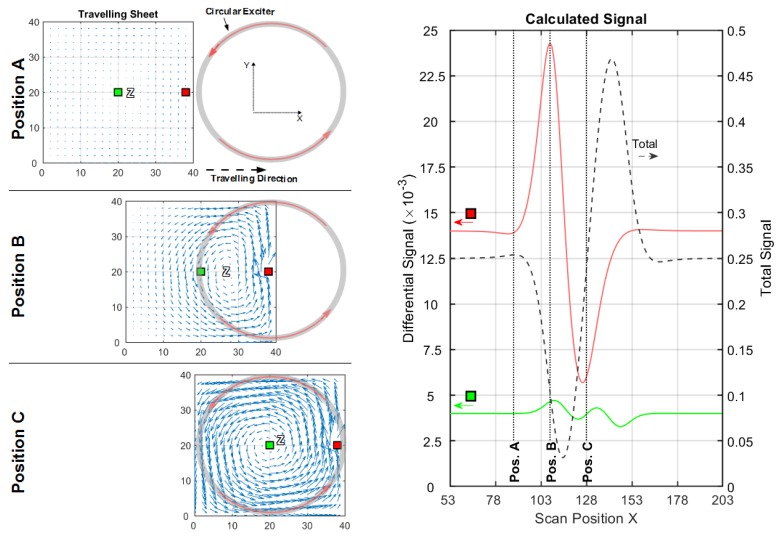
Current and signal calculations with a circular setup (cf. Figure 3a); (**left**) Travelling 2D sheet with local voids (green and red squares) in front of a circular exciter. Blue arrows depict the current direction and intensity. Only three significant x-positions (A, B and C) are shown; (**right**) Total (dotted black line) and differential signals for a void in the center (green line) and for a void at the edge (red line).

**Figure 6 sensors-20-01306-f006:**
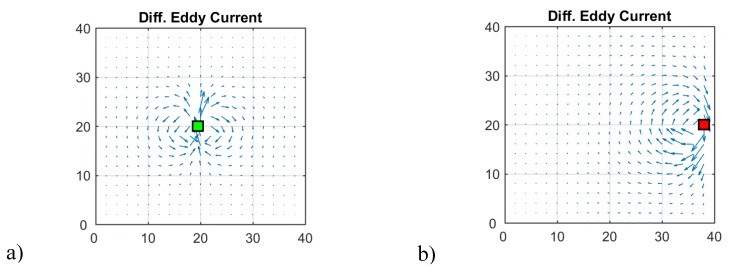
Differential eddy current density fields between a sheet with and without a void at the considered position. (**a**) A void in the middle; (**b**) a void at the edge.

**Figure 7 sensors-20-01306-f007:**
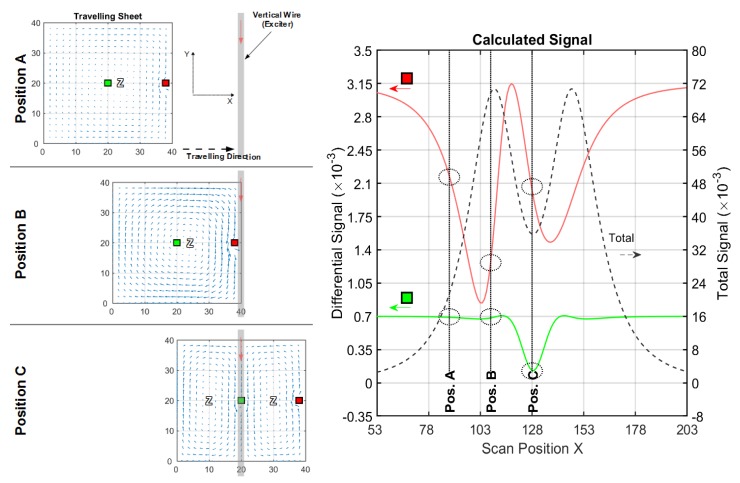
Current and signal calculations with a vertical wire setup (cf. Figure 3b); (**left**) Travelling 2D sheet with local voids (green and red) in front of a vertical wire. Blue arrows depict the current direction and intensity. Only three significant x-positions (A, B and C) are shown; (**right**) Total (dotted black) and differential signals for a void in the center (green) and for a void at the edge (red).

**Figure 8 sensors-20-01306-f008:**
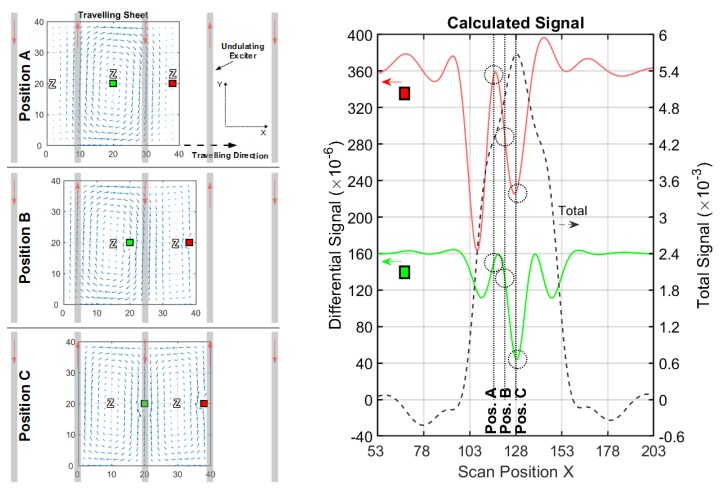
Current and signal calculations with an undulator setup (cf. Figure 3c); (**left**) Travelling 2D sheet with local voids (green and red) in front of an undulator. Blue arrows depict the current direction and intensity. Only three significant x-positions (A, B and C) are shown; (**right**) Total (dotted black) and differential signals for a void in the center (green) and for a void at the edge (red).

**Figure 9 sensors-20-01306-f009:**
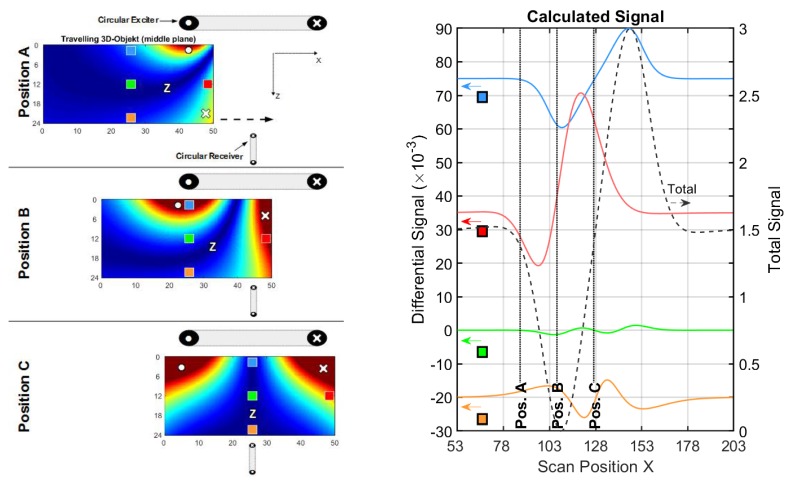
Current and signal calculations with a circular setup (cf. Figure 2 and Figure 3a); (**left**) Eddy current density distribution in the horizontal middle plane of a conducting volume. Three positions (A, B and C) of the travelling cuboid are shown. Local voids at four characteristic positions inside the travelling cuboid (blue, green, orange and red squares) are analyzed; (**right**) Total signal (dotted black) and differential signals (blue, green, orange and red lines) for the four voids.

**Figure 10 sensors-20-01306-f010:**
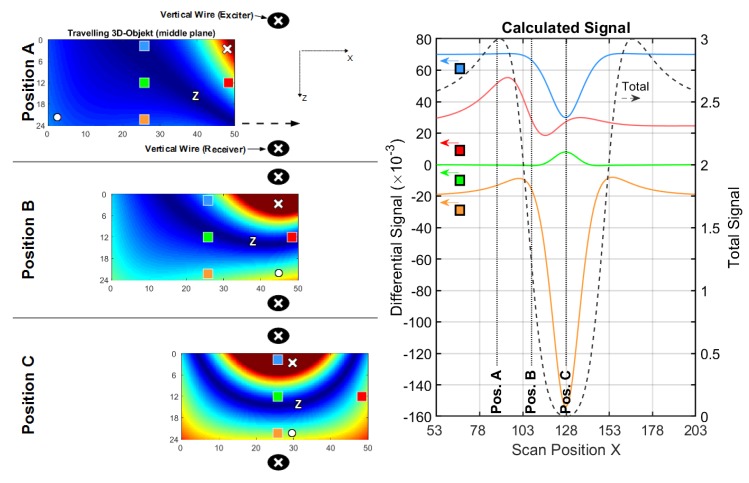
Current and signal calculations with a vertical wire setup (cf. Figure 3b); (**left**) Eddy current density distribution in the horizontal middle plane of a conducting and travelling volume, shown at three positions A, B and C. Local voids at four characteristic positions inside the travelling volume (blue, green, orange and red squares) are analyzed; (**right**) Total signal (dotted black) and differential signals (blue, green, orange and red lines) for the four voids.

**Figure 11 sensors-20-01306-f011:**
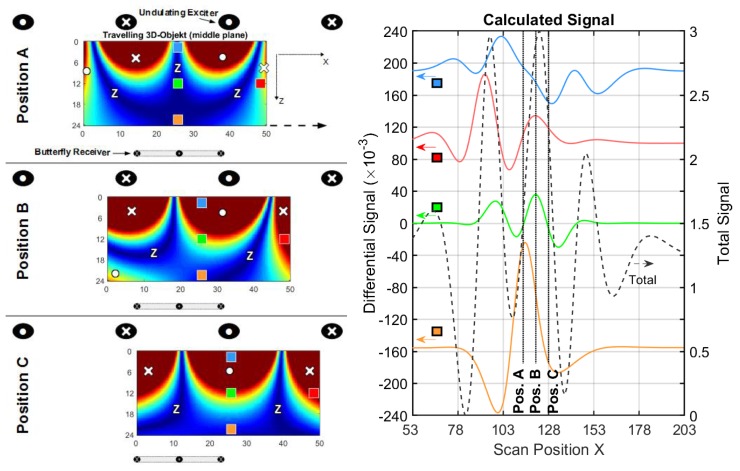
Current and signal calculations with an undulator setup (cf. Figure 3c); (**left**) Eddy current distribution in the horizontal middle plane of a conducting and travelling volume, shown at three characteristic positions A, B and C. Local voids at four characteristic positions inside the travelling volume (blue, green, orange and red squares) are analyzed; (**right**) Total signal (dotted black) and differential signals (blue, green, orange and red lines) for the four voids.

**Figure 12 sensors-20-01306-f012:**
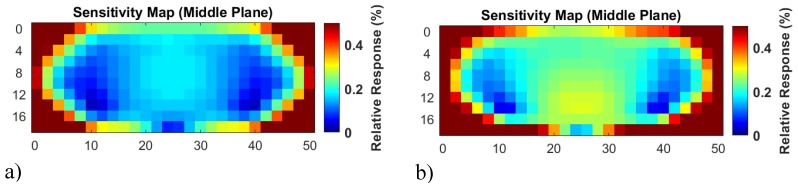
Calculated sensitivity maps for the undulator scanner. (**a**) The response function of the scanner for the horizontal middle plane of a cuboid is condensed to the RMS values for each point; (**b**) high-pass filtered response functions better account for the localization value of the signal.

**Figure 13 sensors-20-01306-f013:**
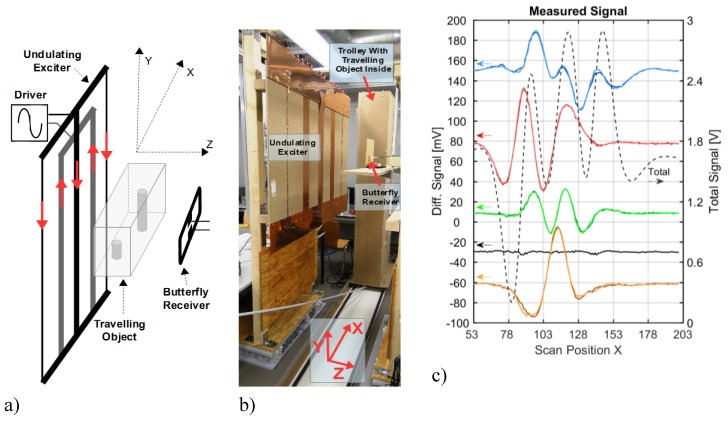
Scanner equipped with an undulating exciter and a gradiometric butterfly receiver. (**a**) Schematic illustration; (**b**) practical realization; (**c**) measured signals from the saline body with a non-conducting sphere (c.f. Figure 2b) at the same positions, as shown in Figure 2c. The dotted black line depicts the total signal of the cuboids; the red lines are the differential signals of the sphere when face-centered at the side wall; the blue and yellowish lines stand for the signature of the sphere towards the exciter and receiver coils (face-centered). The green lines from the center of the volume provide information clearly above the noise floor (black line).

**Figure 14 sensors-20-01306-f014:**
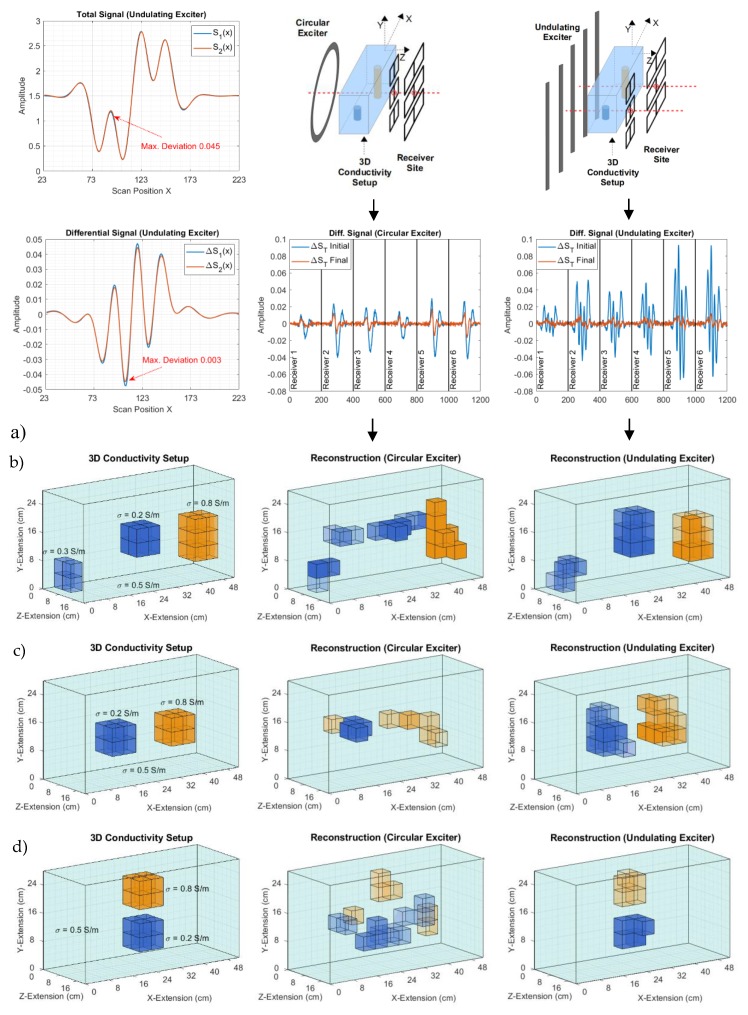
3D reconstructions from the emulated measurements with 50 dB SNR. (**a**) Computed signals from a cuboid with an ideal undulator and an 11-wire undulator (no noise added in this presentation); (**b**) Measurement setup, noise affected signals, and 3D reconstructions are compared for a circular coil and an 11-wire undulator; (**c**) and (**d**) Other perturbations inside the cuboidal volume.

**Figure 15 sensors-20-01306-f015:**
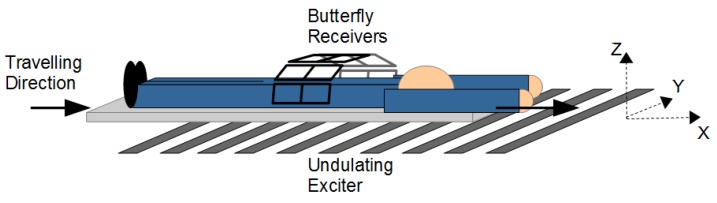
Schematic of a longitudinal scanner setup with a person lying down.

**Table 1 sensors-20-01306-t001:** Main results from the calculated signals, shown in Figure 5.

Total Signal (max. Amplitude)	Edge Diff. Signal (Max. Amplitude)	Center Diff. Signal (Max. Amplitude)	Edge Sensitivity ^1^	Center Sensitivity ^1^ (CAS)
≈0.435	$\approx 18.6\times 10^{-3}$	$\approx 1.436\times 10^{-3}$	≈1/23	≈1/303

^1^ The ratio between the differential signal and the total signal.

**Table 2 sensors-20-01306-t002:** Main results from the calculated signals, shown in Figure 7.

Total Signal (max. amplitude)	Edge Diff. Signal (max. amplitude)	Center Diff. Signal (max. amplitude)	Edge Sensitivity ^1^	CAS ^1^
$\approx 72\times 10^{-3}$	$\approx 2.33\times 10^{-3}$	$\approx 0.57\times 10^{-3}$	≈1/31	≈1/126

^1^ Ratio between the differential signal and the total signal.

**Table 3 sensors-20-01306-t003:** The main results from the calculated signals, shown in Figure 8.

Total Signal (max. amplitude)	Edge Diff. Signal (max. amplitude)	Center Diff. Signal (max. amplitude)	Edge Sensitivity ^1^	CAS ^1^
$\approx 6\times 10^{-3}$	$\approx 235\times 10^{-6}$	$\approx 120\times 10^{-6}$	≈1/26	≈1/50

^1^ The ratio between the differential signal and the total signal.

**Table 4 sensors-20-01306-t004:** Main results from the calculated signals, shown in Figure 9.

Total Signal (max. amplitude)	Edge Diff. Signal (max. amplitude)	Center Diff. Signal (max. amplitude)	Edge Sensitivity ^1^	CAS ^1^
≈2.9	$\approx 50\times 10^{-3}$	$\approx 2.7\times 10^{-3}$	≈1/60	≈1/1074

^1^ Ratio between the differential signal and the total signal.

**Table 5 sensors-20-01306-t005:** Main results from the calculated signals, shown in Figure 10.

Total Signal (max. amplitude)	Edge Diff. Signal (max. amplitude)	Center Diff. Signal (max. amplitude)	Edge Sensitivity ^1^	CAS ^1^
≈3	$\approx 36\times 10^{-3}$	$\approx 8\times 10^{-3}$	≈1/83	≈1/375

^1^ Ratio between the differential signal and the total signal.

**Table 6 sensors-20-01306-t006:** The main results from the calculated signals, shown in Figure 11.

Total Signal (max. amplitude)	Edge Diff. Signal (max. amplitude)	Center Diff. Signal (max. amplitude)	Edge Sensitivity ^1^	CAS ^1^
≈3	$\approx 120\times 10^{-3}$	$\approx 66\times 10^{-3}$	≈1/25	≈1/45

^1^ Ratio between the differential signal and the total signal.

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
