# Peer review of "Three-dimensional Magnetic Induction Tomography: Improved Performance for the Center Regions inside a Low Conductive and Voluminous Body"

_sensors, 2020, doi:10.3390/s20051306_

Round 1
Reviewer 1 Report
The manuscript suggest an improved measurement setup for MIT imaging using undulating exciter coils. It is shown that the central area sensitivity can be increased with the new approach.
The manuscript tackles an important problem in MIT, is clearly written and shows clear improvement in the MIT images. Some minor comments are given for consideration below.
1) IS voluminous a good term?
2) In abstract you ".. with a practical measurement". Perhaps "practical measurements".
3) Mention clearly what sensitivity you are calculating. Sensitivity to conductivity changes, I assume.
4) In equation (2.1), what is the volume V? I thought that the signal to be measured is the integral of the E-field along the measurement coil? Or with Stokes theorem, iωB over the surface covered by the coil. Where does this equation (2.1) come from?
5) I do not quite understand in the case of circular coil, why the measurement coil is perpendicular to the exciter? What if it is parallel? Do you get a bigger signal?
6) Your comparison between the undulating (i do not like this term, by the way) and circular coil is slightly unfair, since your circular coil is large with one loop, but the undulating coil has several loops. You could basically get the same improvement with several circular coils that are excited simultaneously. Same kind of pattern with increased central sensitivity could be achieved, I assume. Comment on this?
7) It seems that you use Landweber methods to solve the conductivity distribution inside the domain. Are you solving a non-linear image reconstruction problem with a non-linear Landweber or something else. This is a bit unclear from your explanation on pages 7 and 8.
Reviewer 2 Report
This review is for the manuscript 'Three-dimensional magnetic induction tomography: improved performance for the center regions inside a low conductive and voluminous body' by M. Klein, D. Erni, and D. Rueter, submitted to Sensors. In this work, the Authors report on a thorough investigation, both numerical and experimental, on how to improve the performance in terms of central area sensitivity (CAS) of magnetic induction tomography (MIT). The direct application discussed by the Authors is to medical imaging.
The first part of the manuscript is dedicated to simulations of different 2D and 3D exciter/detector geometries: planar with circular exciter, planar with vertical wire exciter, and a novel configuration explored in this manuscript, based on a quasi-infinite distribution of vertical wires, with antiparallel current, and regularly spaced ('undulator') with a butterfly receiver. With these configurations, the Authors calculate differential signals, vector potential distributions, noise levels and sensitivity matrices to characterize the various configuration. Results are presented in Figures 7-10 and Tables 1-3 for eddy currents distribution in the case of a 2D conductive sheet moving across the separation between exciter and receiver. The main conclusion in these cases is that the new 'undulator' geometry allows one to obtain a 20 dB improvement in CAS, although at the expense of a reduction in the signal level. Then, the Authors move on to simulate a travelling 3D object, a cuboid of homogeneous conductive material, in the three geometries and configurations described above. Results are presented in Figures 11-14 and Tables 4-6. Results also confirm an increase up to 23 dB in the signal level, in particular from the central regions (as opposed to superficial). To verify the promising performance of the 'undulator' geometry, the Authors also present an experimental demonstration.
Based on their results, the Authors conclude that their 'undulator' geometry can solve - or at least mitigate - the issues of MIT with CAS. They demonstrate gain up to 20 dB, and a net increase of the signal from the central regions with respect to conventional arrangements. Furthermore, the geometry of the 'undulator' allowed them to further simplify image reconstruction and to improve its performance, with specific advantages for the inner regions. They conclude their manuscript proposing a geometry for a longitudinal scanner for a lying patient. The scanner will be formed on a body-wide 'undulating exciter' and an arch of butterfly receivers.
Overall, the manuscript is solid and extremely promising work, which certainly deserved publication in Sensors. It will surely have a relevant impact in the community involved in MIT, and - potentially - on other communities (see my last comment). However, I encourage the Authors to address the following points to further clarify their manuscript and thus increase the impact of their work.
1. In general, sections 2 and 3 are not easy to read, given the (sometimes) large separation between text and figures, and the large number of configurations (simulated and experimental) investigated. The Authors should try to summarize some of the paragraphs, and/or to rearrange their text in a more compact way. I am aware of the difficulty of this task, but sometimes the reader is lost in the text, at the expense of the clarity of the Authors' message.
2. On line 106 the Authors wrote that 'the object travels 256 cm in the x-direction'. This seems in contradiction with Fig. 2b). Is it correct to assume that the 0.27 liter sphere is steady with respect to the saline body? Then, that the saline body is transported during the experiments? If so, the Authors should clarify this in their manuscript.
3. In the case of 'undulator', would it be possible to analyze the contribution of the exciter separately from that of the butterfly receiver? If not possible, the Authors should comment on the combined effect of the two, as opposed to the simple 'undulator' with a more conventional receiver. For example: would this structure work with a different receiver? Would a degradation in performance to be expected?
4. Is there any design strategy to reduce the impact of mechanical noise in the 'undulator' case?
5. Finally, the Authors should mention the recent efforts to perform MIT with atomic magnetometers for medical applications (for example, https://arxiv.org/abs/1805.05743, https://arxiv.org/abs/1905.01661, https://arxiv.org/abs/2002.04943). Although clearly at a different level of technology progress with respect to the configurations explored by the Authors, would the findings of the present work be applicable with these quantum sensors? I believe that the Authors' results may be of interest to that community, so I would suggest the Authors to briefly comment on this, if possible.
Reviewer 3 Report
The paper describing an interesting and challenging issue. In its current form it is hard to follow the story. I suggest the authors clarify points below:
1)Explain in detail the exact application of ihterest. In current form it is hard to see what is the use. Clearly describe the application so it can be best judged weather or not the improvement is useful.
2)Not sure if I understand the sensitivity plots of figure 1. To me they look like dot product of B fields which is not relevant for conductive background when looking for conductivity changes. Plus that how those four coils in figure 1 links to 3D scanning and what is done later on.
3)Also need to clarify if amplitude is measured or phase, and each one of these behave slightly differently.
4)Not sure what is scanning person in figure 17 is showing. What is the purpose and why we are scanning whole body by moving the person?
Round 2
Reviewer 3 Report
The paper now does better on clarifying my comments.